# Changes in leisure-time physical activity during the adult life span and relations to cardiovascular risk factors—Results from multiple Swedish studies

**Lars Lind[1]***, **Björn Zethelius[2]**, **Eva Lindberg[3]**, **Nancy L. Pedersen[4]**, **Liisa Byberg[5]**

**1** Department of Medical Sciences, Uppsala University, Uppsala, Sweden, **2** Department of Public Health/Geriatrics, Uppsala University, Uppsala, Sweden, **3** Department of Medical Sciences/Respiratory, Allergy and Sleep Medicine, Uppsala University, Uppsala, Sweden, **4** Department of Medical Epidemiology and Biostatistics, Karolinska Institute, Stockholm, Sweden, **5** Department of Surgical Sciences, Uppsala University, Uppsala, Sweden

* lars.lind@medsci.uu.se

**Data Availability Statement:** Due to Swedish laws on personal integrity and health data, as well as the Ethics Committee, we are not allowed to make any

## Abstract

### Objective

To evaluate how self-reported leisure-time physical activity (PA) changes during the adult life span, and to study how PA is related to cardiovascular risk factors using longitudinal studies.

### Methods

Several Swedish population-based longitudinal studies were used in the present study (PIVUS, ULSAM, SHE, and SHM, ranging from hundreds to 30,000 participants) to represent information across the adult life span in both sexes. Also, two cross-sectional studies were used as comparison (EpiHealth, LifeGene). PA was assessed by questionnaires on a four or five-level scale.

### Results

Taking results from several samples into account, an increase in PA from middle-age up to 70 years was found in males, but not in females. Following age 70, a decline in PA was seen. Young adults reported both a higher proportion of sedentary behavior and a higher proportion high PA than the elderly. Females generally reported a lower PA at all ages.

PA was mainly associated with serum triglycerides and HDL-cholesterol, but also weaker relationships with fasting glucose, blood pressure and BMI were found. These relationships were generally less strong in elderly subjects.

### Conclusion

Using data from multiple longitudinal samples the development of PA over the adult life span could be described in detail and the relationships between PA and cardiovascular risk

data, including health variables, open to the public even if made anonymous. The data could be shared with other researchers after a request to the steering committee (karl.michaelsson@surgsci.uu. se).

**Funding:** The authors received no specific funding for this work.

**Competing interests:** The authors have declared that no competing interests exist.

factors were portrayed. In general, a higher or increased physical activity over time was associated with a more beneficial cardiovascular risk factor profile, especially lipid levels.

## Introduction

Based on observational data, there is a general agreement that a high leisure time physical activity (PA) is beneficial in terms of future CVD [1, 2]. The same is considered for a high cardiorespiratory fitness, a condition most often accompanying a high PA [3, 4]. However, high cardiorespiratory fitness is also dependent on biological and genetic factors that may be independent of physical activity [5]. In a large study including twins in different European countries, the heritability of PA in males and females was similar and ranged from 48% to 71% [6]. Such high heritability was confirmed in another twin study [7]. The estimated heritability of CVDs, like coronary heart disease, is somewhat lower (30–60%) [8].

Two studies have used the Mendelian randomization (MR) approach to see if genes linked to PA also are linked to CVD, thereby suggesting a causal effect of PA on CVD. One of these studies showed a suggestive causal genetic effect of PA on coronary heart disease [9], but not stroke, while no such associations were found in another study using MR [10]. It should however be remembered that the number and strength of the SNPs used as genetic instrument for PA is quite low and therefore the risk of false negative findings are high.

Long-term randomized trials with increased PA does not exist, but a community-based intervention study performed in the primary care setting over 9 months showed an improvement in blood pressure and lipids after 9 months, but also a significant reduction in incident CVD after 2 years [11].

Generally, the amount of high PA declines with ageing, while the amount of sedentary behavior increases with age [12–15]. This pattern has been reported to be more pronounced in women compared to men [16]. However, most of those studies are cross-sectional and only a few longitudinal studies exist that are addressing the issue of change in PA over longer time periods [17, 18]. Thus, there is a need for more longitudinal studies covering the adult lifespan. Since it is hard to cover the whole adult lifespan in a single study, we have in this paper used multiple studies to reflect different parts of the lifespan in both men and women.

The positive impact of a high PA on CVD might well be due to positive effects on traditional risk factors for CVD, such as a better glucose control [19], improved lipid levels [20], and a lower blood pressure [21], as documented in intervention studies. However, since the impact of these risk factors on future risk for CVD vary with age [22], we hypothesized that also the relationship between PA and CVD risk factors varied in strength during the lifespan. Ideally, also in this case longitudinal data should be used in which the relationship between PA and CVD risk factors should be studied in the same sample at different ages.

Thus, the primary objective of the study was to evaluate how PA levels change over time and how this is dependent on age and sex. The second objective was to study how PA is related to traditional risk factors for cardiovascular disease, such as blood pressure, lipids, diabetes, obesity, and smoking in different age group and in men and women. For both of these objectives, we used several longitudinal study samples to cover both sexes and most of the adult lifespan. Thus, the major contribution of the present study to the already existing knowledge in this field is that we are able to study these two objectives in the longitudinal setting using several study samples.

As a complement to the longitudinal data, we also report results from a large cross-sectional study covering the age span from 20 to 75 years.

## Methods

### Samples

ULSAM (Uppsala Longitudinal Study of Adult Men) is a population-based study of all men aged 50 living in Uppsala during the years 1970–74. The participation rate was 82%. Re-examinations have been performed at ages 60, 70, 77, 82, 88, and 93. The two latest investigations are not used in the present analysis. Details on the study sample is given in [23] (www.pubcare.uu.se/ULSAM). Data on PA was present in 2291 of the subjects at baseline. See Table 1 for n at the other investigations.

PIVUS (Prospective Investigation of Vasculature in Uppsala Seniors) is a population-based study of men and women aged 70 in Uppsala during the years 2001–2004. The participation rate was 50%. Re-examinations have been performed at ages 75 and 80. Details on the study sample is given in [24]. Data on PA was present in 1013 of the subjects at baseline. See Table 1 for n at the other investigations.

EpiHealth (Epidemiology for Health) is a population-based study of men and women aged 45–75 in the cities of Uppsala and Malmö during the years 2011–2018. The participation rate was 20%. No re-examinations have been performed. Details on the study sample is given in [25, 26]. Data on PA was present in 24,703 of the subjects.

LifeGene is population-based study of men and women from the newborn to mid-age in the cities of Stockholm, Umeå, and Alingsås during the years 2009–2018. No re-examinations have been performed. Details on the study sample is given in [27]. Data on PA was present in 21,759 of the subjects.

SHE (Sleep and Health in Women) is a population-based study of women aged 20–99 years in Uppsala in 2000. The participation rate was 71.6%. Re-examination was performed in 2010. Details on the study sample is given in [28]. Data on PA was present in 6955 of the subjects at baseline. See Table 1 for n at the other investigations.

SHM (Sleep and Health in Men) is a population-based study of men aged 30–69 years and living in Uppsala in 1984. The participation rate was 79.6%. Re-examinations were performed in 1994 and 2007. Details on the study sample is given in [29]. Data on PA was present in 2626 of the subjects at baseline. See Table 1 for n at the other investigations.

The participants of each study gave written informed consent and the studies were either approved by the Regional Ethics Committee in Uppsala (ULSAM, PIVUS, EpiHealth, SHE, and SHM) or Stockholm (LifeGene), Sweden.

### PA assessment

In all studies, a questionnaire was given to the participants who answered questions regarding leisure time PA. The questions were different across the studies (details given in S1 Table), but the results were presented in 4 (5 in EpiHealth and LifeGene) PA categories with 1 denoting a sedentary behavior and 4 (5 in EpiHealth and LifeGene) denoting an athletic lifestyle.

### Risk factors

Blood samples were drawn in all studies after an overnight fast, except in EpiHealth in which only 6 hours of fasting was required. Plasma glucose, serum triglycerides, LDL-, and HDL-cholesterol were measured by standard techniques, see the references above for details. Blood pressure was measured in the supine position in ULSAM and PIVUS and in the sitting position in EpiHealth.

Smoking status, education level, and medications were obtained using questionnaires.

**Table 1. Description of studied variables in the different samples.**

**ULSAM**

| Description | Population-based study in Uppsala including men aged 50. Baseline collected in 1970–1974. Five examination cycles. | | | | | | | | | |
|---|---|---|---|---|---|---|---|---|---|---|
| Examinations | Age 50 | | Age 60 | | Age 70 | | Age 77 | | Age 82 | |
| | N | Mean (SD)/ proportion | N | Mean (SD)/ proportion | N | Mean (SD)/ proportion | N | Mean (SD)/ proportion | N | Mean (SD)/ proportion |
| SBP (mmHg) | 2291 | 133.05 (18.01) | 1836 | 142.53 (19.63) | 1214 | 146.77 (18.5) | 834 | 150.52 (19.99) | 525 | 145.06 (17.47) |
| Triglycerides (mmol/l) | 2292 | 1.93 (1.24) | 1836 | 1.66 (.7) | 1214 | 1.45 (.77) | 834 | 1.38 (.69) | 525 | 1.39 (.68) |
| HDL-cholesterol (mmol/l) | 2125 | 1.36 (.38) | 1742 | 1.28 (.24) | 1213 | 1.28 (.35) | 832 | 1.31 (.32) | 524 | 1.2 (.29) |
| LDL-cholesterol (mmol/l) | 2123 | 5.26 (1.19) | 1740 | 4.43 (.66) | 1213 | 3.89 (.9) | 832 | 3.47 (.85) | 524 | 3.39 (.84) |
| BMI (kg/m$^2$) | 2292 | 25 (3.19) | 1836 | 25.48 (3.28) | 1214 | 26.28 (3.42) | 834 | 26.3 (3.44) | 525 | 26.09 (3.43) |
| Glucose (mmol/l) | 2290 | 5.58 (.97) | 1836 | 5.58 (1.43) | 1214 | 5.77 (1.47) | 834 | 5.88 (1.38) | 525 | 5.94 (1.24) |
| Antihypertensive medication (%) | 2322 | 4 | 1860 | 19 | 1210 | 35 | 782 | 42 | 477 | 54 |
| Lipid lowering medication (%) | 2322 | 1 | 1860 | 6 | 1151 | 9 | 782 | 17 | 477 | 21 |
| Antidiabetic medication (%) | 2322 | 1 | 1855 | 2 | 1151 | 6 | 782 | 9 | 477 | 10 |
| MetS (%) | 2123 | 12 | 1737 | 12 | 1145 | 15 | 775 | 15 | 474 | 20 |
| MetS components | 2123 | 1.35 (.97) | 1737 | 1.39 (.9) | 1145 | 1.52 (.94) | 775 | 1.57 (.92) | 474 | 1.71 (.96) |

**PIVUS**

| Description | Population-based study in Uppsala including men and women aged 70. Baseline collected in 2001–2004. Three examination cycles. | | | | | | | | | |
|---|---|---|---|---|---|---|---|---|---|---|
| Examinations | Age 70 | | Age 75 | | Age 80 | | | | | |
| | N | Mean (SD)/ proportion | N | Mean (SD)/ proportion | N | Mean (SD)/ proportion | | | | |
| Sex (% females) | 1016 | 50 | 826 | 50 | 606 | 50 | | | | |
| SBP (mmHg) | 1012 | 149.63 (22.68) | 825 | 148.75 (19.44) | 607 | 146.86 (19.47) | | | | |
| Triglycerides (mmol/l) | 1013 | 1.28 (.6) | 826 | 1.39 (.66) | 605 | 1.24 (.59) | | | | |
| HDL-cholesterol (mmol/l) | 1013 | 1.51 (.43) | 825 | 1.49 (.46) | 606 | 1.38 (.39) | | | | |
| LDL-cholesterol (mmol/l) | 1011 | 3.38 (.88) | 825 | 3.37 (.94) | 605 | 3.2 (.9) | | | | |
| BMI (kg/m$^2$) | 1016 | 27.03 (4.33) | 826 | 26.87 (4.38) | 604 | 26.91 (4.52) | | | | |
| Glucose (mmol/l) | 1013 | 5.34 (1.61) | 826 | 5.24 (1.47) | 599 | 5.28 (1.38) | | | | |
| Antihypertensive medication (%) | 1013 | 31 | 826 | 50 | 606 | 61 | | | | |
| Lipid lowering medication (%) | 1016 | 15 | 826 | 25 | 606 | 30 | | | | |
| Antidiabetic medication (%) | 1013 | 6 | 943 | 10 | 606 | 11 | | | | |
| MetS | 982 | 21 | 810 | 30 | 576 | 30 | | | | |
| MetS components | 982 | 1.68 (1.11) | 810 | 1.98 (1.18) | 576 | 2.01 (1.13) | | | | |

**EpiHealth**

| Description | Population-based study in Uppsala/Malmö including men and women aged 45–75. Data collected in 2010–2018. | | | | | | | | | |
|---|---|---|---|---|---|---|---|---|---|---|
| Age-group | 45–54 | | 55–64 | | 65–75 | | | | | |
| | N | Mean (SD)/ proportion | N | Mean (SD)/ proportion | N | Mean (SD)/ proportion | | | | |
| Age | 7313 | 50.07 (3) | 8231 | 60.23 (2.9) | 9806 | 69.65 (3.02) | | | | |
| Sex (% females) | 7313 | 61 | 8231 | 58 | 9806 | 52 | | | | |
| SBP (mmHg) | 7311 | 126.88 (15.05) | 8226 | 134.56 (16.51) | 9803 | 141.3 (17.58) | | | | |
| Triglycerides (mmol/l) | 7213 | 1.17 (.76) | 8139 | 1.27 (.75) | 9736 | 1.28 (.63) | | | | |
| HDL-cholesterol (mmol/l) | 7214 | 1.53 (.39) | 8139 | 1.57 (.41) | 9735 | 1.56 (.4) | | | | |
| LDL-cholesterol (mmol/l) | 7210 | 3.59 (.9) | 8139 | 3.83 (.95) | 9731 | 3.75 (1.01) | | | | |

*(Continued)*

**Table 1.** (Continued)

| BMI (kg/m²) | 7252 | 26.09 (4.16) | 8066 | 26.39 (4.04) | 9385 | 26.49 (3.95) | | | | |
|---|---|---|---|---|---|---|---|---|---|---|
| Glucose (mmol/l) | 7212 | 5.75 (.81) | 8136 | 6 (1.02) | 9742 | 6.16 (1.12) | | | | |
| Antihypertensive medication (%) | 7313 | 8 | 8230 | 21 | 9804 | 35 | | | | |
| Lipid lowering medication (%) | 7313 | 3 | 8230 | 10 | 9804 | .19 | | | | |
| Antidiabetic medication (%) | 7313 | 1 | 8230 | 3 | 9804 | 5 | | | | |
| MetS | 7203 | 16 | 8124 | 25 | 9729 | 29 | | | | |
| MetScomponents | 7203 | 1.32 (1.19) | 8124 | 1.73 (1.22) | 9729 | 1.94 (1.15) | | | | |
| **LifeGene** | | | | | | | | | | |
| Description | Population-based study in Stockholm/Umeå/Alingsås including men and women aged 20–50. Data collected in 2010–2018. | | | | | | | | | |
| | N | Mean (SD)/ proportion | | | | | | | | |
| Age | 21759 | 32.40 (7.2) | | | | | | | | |
| Sex (% female) | 21759 | 58 | | | | | | | | |
| **SHE** | | | | | | | | | | |
| Description | Population-based study in Uppsala including women aged 20–99. Baseline collected in 2001. | | | | | | | | | |
| Year | 2001 | | 2010 | | | | | | | |
| | N | Mean (SD)/ proportion | N | Mean (SD)/ proportion | | | | | | |
| Age | 6955 | 45.19 (17.4) | 5145 | 53.11 (15.2) | | | | | | |
| **SHM** | | | | | | | | | | |
| Description | Population-based study in Uppsala including men aged 40–79. Baseline collected in 1994. | | | | | | | | | |
| Year | 1994 | | 2007 | | | | | | | |
| | N | Mean (SD)/ proportion | N | Mean (SD)/ proportion | | | | | | |
| Age | 2626 | 55.01 (11.0) | 2006 | 65.10 (9.1) | | | | | | |

SBP, Systolic blood pressure; BMI, Body mass index; MetS, Metabolic syndrome.

The metabolic syndrome (MetS) was defined according to the consensus criteria [30], being a slight modification of the NCEP criteria. The number of the five different components were calculated.

## Statistical analyses

**Changes in PA over time/with age.** In all longitudinal samples the change in PA categories over time was assessed by mixed models for an ordinal outcome (command xtologit). To evaluate if age or sex were related to the change over time in PA, interaction terms between time and age or time and sex were included in the models. All data from all examinations were used in the longitudinal analyses. In ULSAM and PIVUS, the analyses included the confounders education and smoking status (updated for each examination), and sex in PIVUS (ULSAM consisted of men only).

In the cross-sectional analysis, data from LifeGene and EpiHealth were merged and ordinal logistic regression (command ologit) was used to relate age and sex to the PA categories. An interaction term between age and sex was also included in the model, as well as the confounders alcohol intake, smoking status, and education level. For the corresponding figure, the sample was divided into six age-groups (20–29, 30–39, 40–49, 50–59, 60–69 and 70–75).

For the calculations of trajectories in PIVUS and ULSAM, we calculated group-based trajectories using a finite mixed model (command traj). Maximum likelihood is used for the estimation of the model parameters. The maximization is performed using a general quasi-Newton procedure.

**Cross-sectional relationships between PA and CV risk factors.** Fasting glucose and triglycerides were log-transformed to achieve normal distributions.

A trend test for PA used as a continuous variable vs six risk factors (one by one) were performed by linear regression (command regress) for each examination in PIVUS and ULSAM.

EpiHealth was divided into three age-groups (45–54, 55–64, and 65–75), and the analyses were performed in each age-group.

All analyses included the confounders education, smoking status, antihypertensive medication, antidiabetic medication, lipid-lowering drugs, and sex (in PIVUS and EpiHealth).

In the ULSAM study, we evaluated the role of BMI as a mediator in the PA vs risk factor relationship using structural equation models (SEM) using a maximum likelihood method. We then used data for PA and BMI from age 50, while for the outcomes HDL and triglyceride (TG) data from age 60 was used.

**Longitudinal relationships between PA and CV risk factors.** In PIVUS and ULSAM, the relationships between the change in PA and the change in six risk factors (one by one) were performed using mixed models with a random intercept (command xtmixed). The analyses included the confounders education, smoking status, antihypertensive medication, antidiabetic medication, lipid-lowering drugs (updated for each examination), sex (in PIVUS), and the baseline value of PA.

We also evaluated if the trajectories for PA identified in PIVUS and ULSAM were related to the risk factors during the follow-up period. For this task, we used mixed models (command xtmixed) where the first trajectory (see Fig 3) was treated as the reference group and the other trajectories for PA were evaluated vs this reference group.

**Cross-sectional relationships between PA and MetS.** A trend test for PA vs MetS (binary) or the number of MetS components were performed by ordinal logistic regression (command ologit) for each examination in PIVUS and ULSAM. EpiHealth was divided in three age-groups (45–54, 55–64, and 65–75), and the analyses were performed in each age-group. The log odds of the beta coefficients are given.

All analyses included the confounders education, smoking status, and sex (in PIVUS and EpiHealth).

**Longitudinal relationships between PA and MetS.** In PIVUS and ULSAM the relationships between the change in PA and the change in occurrence of MetS or number of MetS components were performed using mixed models for ordinal logistic regression (command xtologit). The analyses included the confounders education, smoking status (updated for each examination), and sex (in PIVUS), and the baseline value of PA.

Calculations were performed using STATA16.1 (Stata inc, College Station, TX, USA).

## Results

Characteristics of the populations are shown in Table 1, and the timing of the data collection in the populations is shown in Fig 1.

### Changes in PA with age

The changes in the proportions of the PA-categories over time in ULSAM are given in Fig 2A. A slight increase in the PA-activity was seen over time (p = 0.00024) when adjusted for

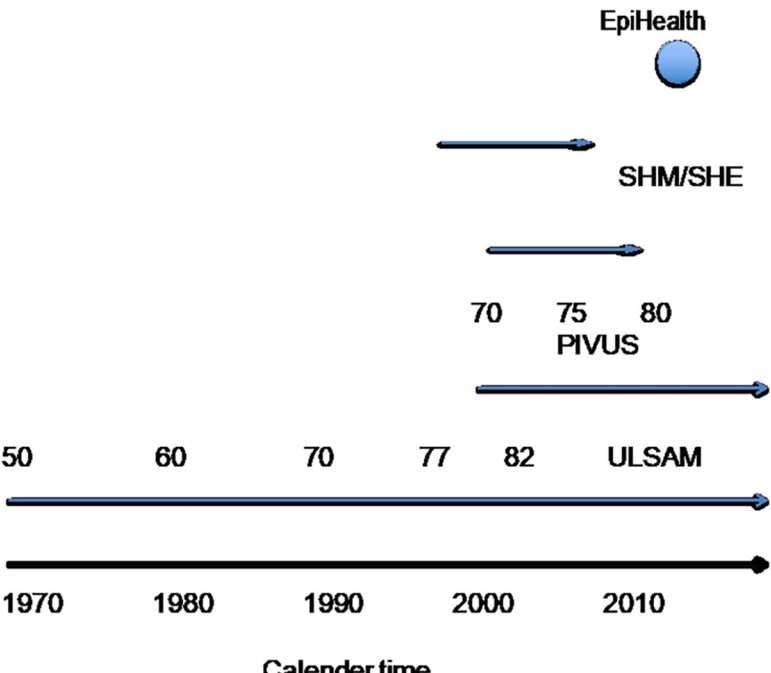

**Fig 1. Overview of the different studies in relation to calendar time.**

smoking and education level. This increase was most pronounced when comparing the 50 to 70-year time-span (p = 5.5e-11). Thereafter, a decline in PA was seen.

The changes in the proportions of the PA-categories over time in PIVUS are given in Fig 2B. The PA level declined over time in the PIVUS study (p = 1.3e-06) when adjusted for sex, smoking, and education level. No significant differences between men and women were seen regarding PA (p = 0.32). The interaction term between time and sex was not significant (p = 0.46).

The changes in the proportions of the PA-categories over time in SHE are given in Fig 2C. The PA level declined over time in the SHE study (p = 1.2e-05) when adjusted for age. The interaction term between time and age was not significant (p = 0.84).

The changes in the proportions of the PA-categories over time in SHM are given in Fig 2D.

A highly significant interaction was seen between time and age (p = 2.6e-08). In the two youngest age-groups, the PA level increased over time (p = 6.4e-03 and p = 3.4e-04, respectively), while on the contrary a reduction in PA over time was seen in the two oldest groups (p = 3.3e-03 and p = 3.6e-05, respectively).

The relationships between age and PA in the cross-sectional analysis in EpiHealth/LifeGene are given in Fig 2E. The physical activity level declined with age (p = 1.0e-11) and was lower in women then in men (p = 3.7e-09) following adjustment for alcohol intake, smoking, and education level. No interaction between age and sex was seen regarding PA (p = 0.54).

The results in the SHM study were stratified by age-group at the initial investigation.

## Individual changes in PA over time

In the ULSAM and PIVUS studies, trajectories for PA based on individual changes were calculated (see Fig 3).

In ULSAM, four significant trajectories were identified (p<0.001 for all). The most common of those showed subjects being moderately active at age 50 and maintaining this level

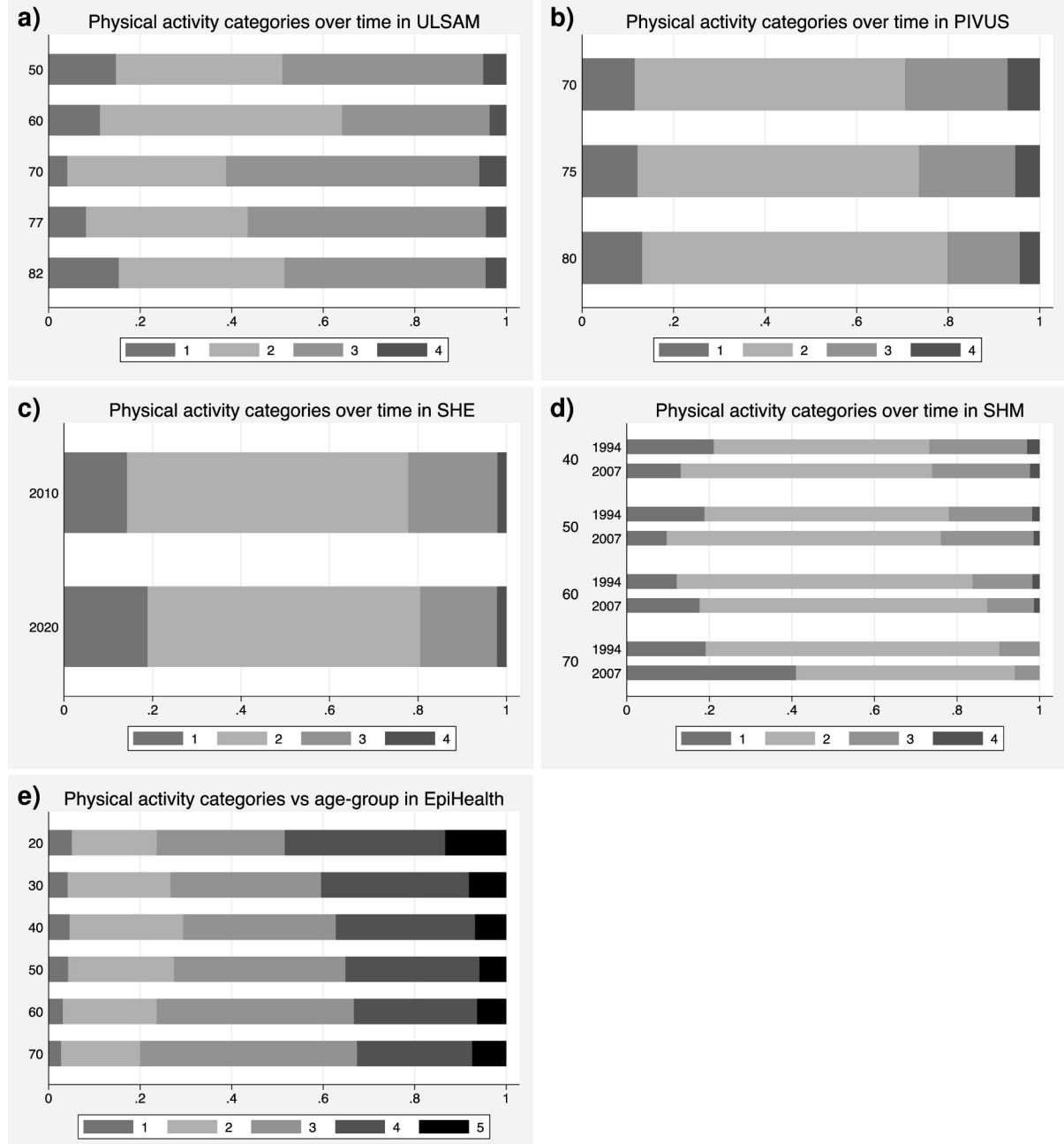

**Fig 2. Longitudinal change in physical activity (PA) categories over time in a) ULSAM, b) PIVUS, c) SHE, and d) SHM and cross-sectional relationships between PA categories and age-groups in e) the EpiHealth study.**

during the follow-up. Another group with a similar moderately active behavior at age 59, declined in activity over time. One sedentary group at age 50 increased their activity up to age 70. A small group showed a high PA at age 50, with only a minor decline by age.

In PIVUS, only three significant trajectories were identified (p<0.001 for all). The vast majority belonged to a group with moderate PA at age 70 and thereafter a minor decline during the 10-year follow-up. A smaller group was sedentary throughout this period, and another group showed a rather high PA at age 70 with a slight decline with ageing.

a)

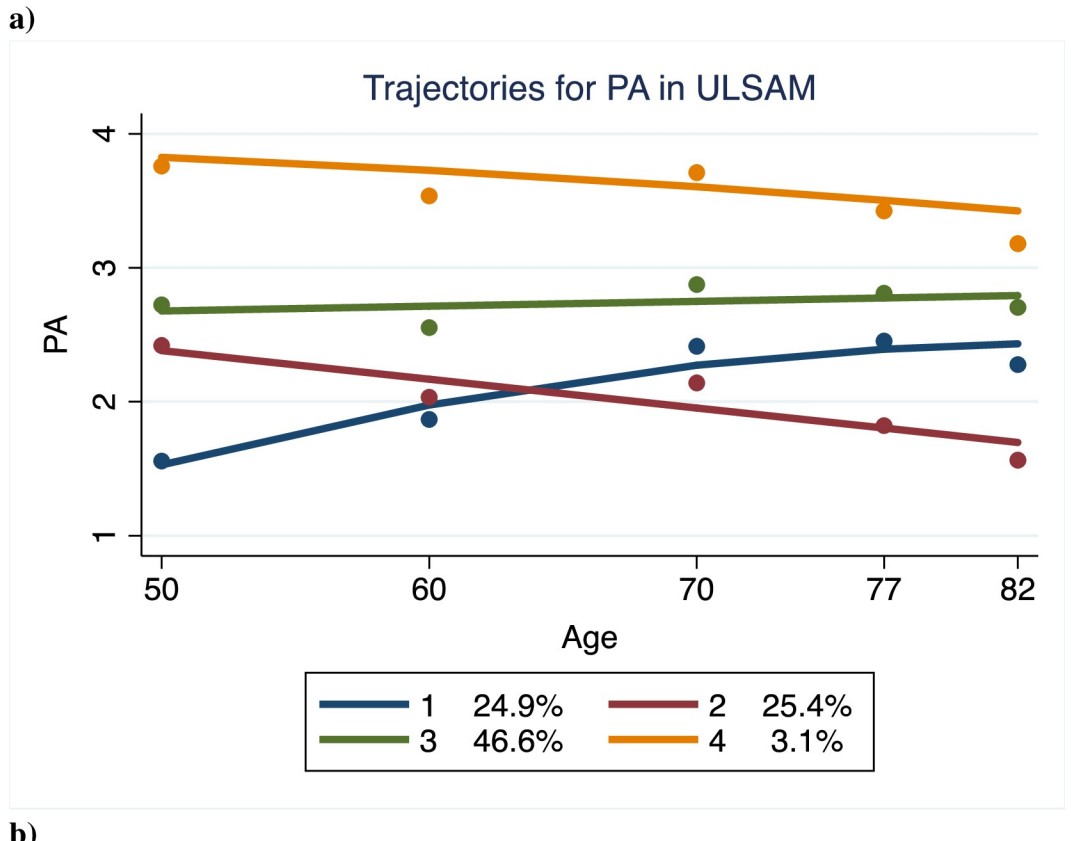

b)

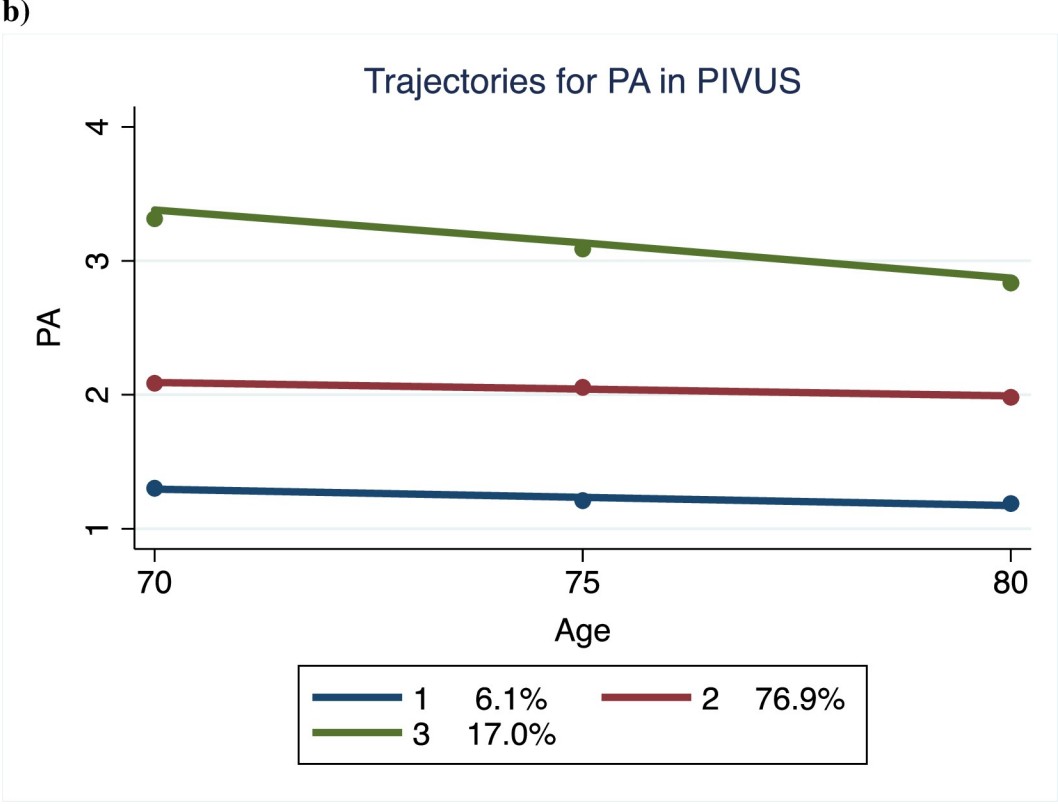

**Fig 3. Trajectories for physical activity (PA) in a) ULSAM and b) PIVUS. The proportion of subjects in each trajectory is given.**

## Cross-sectional relationships between PA and CV risk factors

As can be seen in Fig 4, the regression coefficients for the relationships between PA and CV risk factors were in the same order in all three studies (betas ±0.15–0.20 per higher PA level for the strongest relationships).

In EpiHealth, a fairly homogenous picture was seen with highly significant inverse relationships between PA and glucose, BMI, and triglycerides for each of the age-groups, while a direct association was seen between PA and HDL-cholesterol. These relationships were generally less strong in the highest age-group, although still highly significant. PA was also inversely associated with LDL-cholesterol and systolic blood pressure (SBP), but in this case a major drop in the strength of association was seen in the highest age-group, not being significant in this group.

In ULSAM, similar relationships as described above were seen, with inverse relationships between PA and glucose, BMI, and triglycerides, and a positive relationship with HDL-cholesterol. However, in this case, the relationships at the different ages were more heterogeneous, and associations with LDL-cholesterol and SBP were substantially different, especially when the relationships at high age were compared with those seen at age 50.

In PIVUS, PA was consistently and inversely associated with triglycerides and BMI at all three examinations.

The details from the relationships described above are given in S2 Table.

## Analyses with BMI as a confounder

In ULSAM, including BMI as a confounder in the relationship between PA and triglycerides (TG, at the ln-scale) at age 50 years, resulted in a reduction in the regression coefficient from -0.056 to -0.048 (p<0.0001 for both models).

Including BMI as a confounder in the relationship between PA and HDL at age 50 years in ULSAM, resulted in a reduction in the regression coefficient from 0.025 (p = 0.025) to 0.019 (p = 0.072).

## Analyses with BMI as a mediator

Assuming a causal relationship being: PA at age 50 ->BMI at age 50 ->TG at age 60, showed that BMI was a significant mediator (p = 0.048) accounting for 20% of the total effect of PA on TG. If we, on the contrary, postulated a causal relationship being: BMI at age 50 ->PA at age 50 ->TG at age 60, PA was not a significant mediator of the effect of BMI on TG (p = 0.13).

Assuming a causal relationship being: PA at age 50 ->BMI at age 50 ->HDL at age 60 showed that BMI was a significant mediator (p = 0.049) accounting for 18% of the total effect of PA on HDL.

If we, on the contrary, postulated a causal relationship being: BMI at age 50 ->PA at age 50 ->HDL at age 60, PA was not a significant mediator of the effect of BMI on HDL (p = 0.18).

## Longitudinal relationships between PA and CV risk factors

In ULSAM, the change in PA between 50 and 82 years was related to the change in 6 traditional risk factors (evaluated one by one), with smoking, antihypertensive treatment, statin use, and antidiabetic treatment and education level as confounders. The change in PA was significantly related to the changes in serum triglycerides and HDL-cholesterol. An increase in PA over time was related to a reduction in triglycerides (negative beta coefficient) and an increase in HDL-cholesterol (positive beta coefficient, see Table 2).

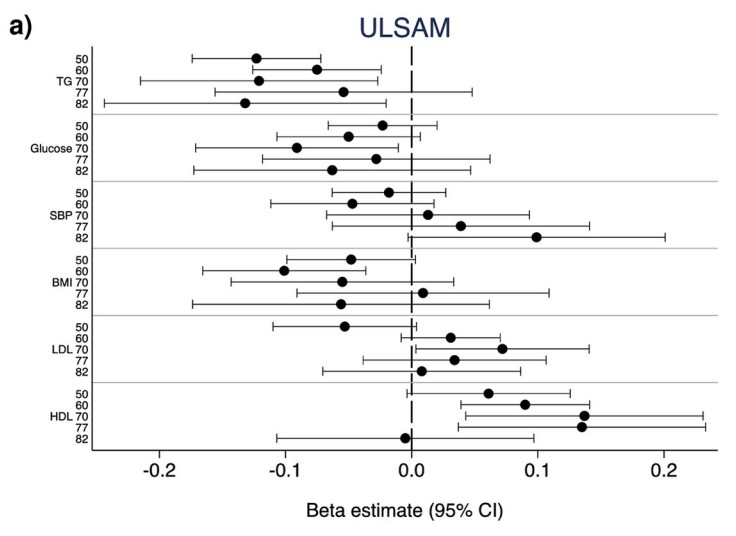

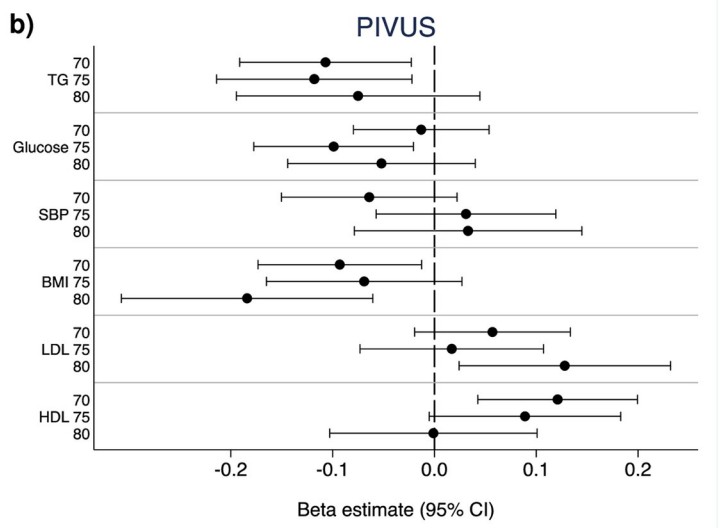

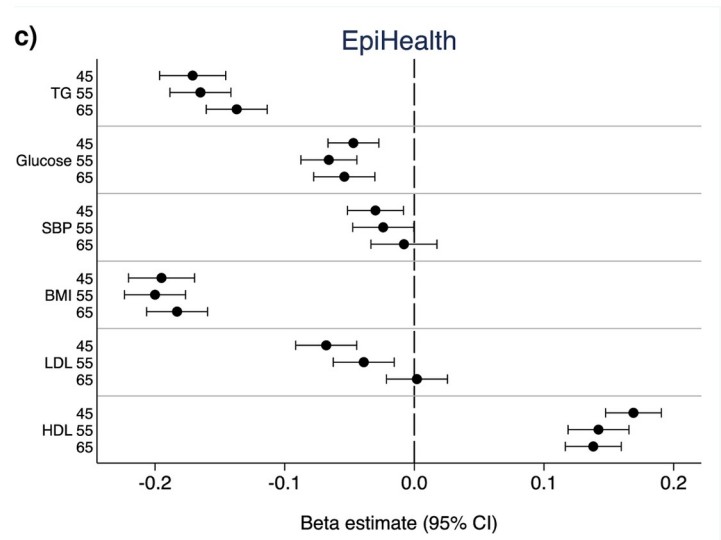

**Fig 4. Relationships between physical activity and different cardiovascular risk factors at different examinations in the a) ULSAM and b) PIVUS studies, and in different age-groups in the c) EpiHealth study.** The regression coefficient (Beta) and 95%CI vs PA is given. Details are given in S1 Table. TG, Triglycerides; SBP, Systolic blood pressure; BMI, Body mass index.

Similar results were seen when only the 50 to 70-year time-span was used in the analyses, but in this case also the change in systolic blood pressure tended to be related to the change in PA in an inverse fashion (p = 0.034).

When the four PA trajectories described in Fig 3 were related to the risk factors, compared with trajectory 1 (those increasing their PA from a sedentary level), trajectories 3 and 4 (both being at higher levels of PA over time) showed significantly (p<0.05) lower levels of TG and BMI, but higher levels of HDL during the time span evaluated. Regarding SBP, lower levels were seen only for trajectory 4. No significant differences between the PA trajectories were seen for fasting glucose or LDL.

In PIVUS, the change in PA between 70 and 80 years was significantly related to the changes in glucose and BMI (negative) and HDL-cholesterol (positive). The inverse relationships vs the change in triglycerides was not significant (p = 0.080, see Table 2).

When the three PA trajectories described in Fig 3 were related to the risk factors, compared with trajectory 1 (constant low activity), trajectories 2 and 3 showed significantly (p<0.05) lower levels of BMI, but higher levels of HDL during the time-span evaluated. Regarding glucose and TG, lower levels were seen only for trajectory 3. No significant differences between the PA trajectories were seen for SBP or LDL.

## Cross-sectional relationships between PA and MetS

As can be seen in Fig 5, the regression coefficients (log odds) for the relationships between PA and MetS and number of MetS components were in the same order in all three studies (beta coefficients -0.50 to -0.35 for the strongest relationships).

**Table 2. Relationships between changes in cardiovascular risk factors and change in physical activity from age 50 to age 82 in ULSAM, and between age 70 and 80 in PIVUS.**

| ULSAM | | | |
|---|---|---|---|
| Variable | Beta | SE | p-value |
| TG | -.081 | .014 | 2.13e-08 |
| Glucose | -.007 | .016 | .67 |
| SBP | .019 | .017 | .26 |
| BMI | .001 | .011 | .95 |
| LDL- cholesterol | -.014 | .013 | .26 |
| HDL- cholesterol | .072 | .014 | 1.36e-07 |
| **PIVUS** | | | |
| Variable | Beta | SE | p-value |
| TG | -.051 | .028 | .080 |
| Glucose | -.068 | .026 | .0085 |
| SBP | .033 | .032 | .298 |
| BMI | -.046 | .015 | .0018 |
| LDL-cholesterol | .021 | .027 | .43 |
| HDL- cholesterol | .044 | .021 | .042 |

TG, Triglycerides; SBP, Systolic blood pressure; BMI, Body mass index.

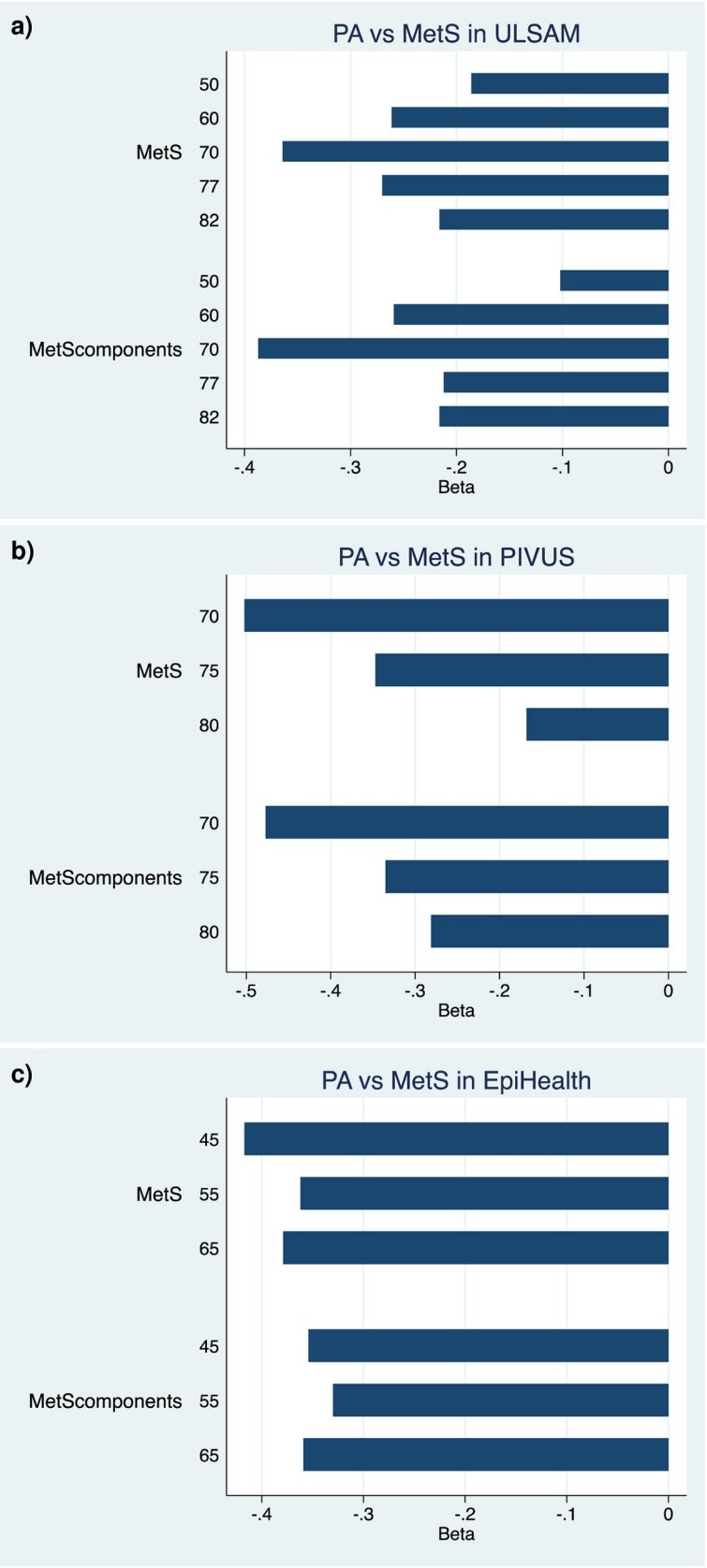

**Fig 5. Relationships between physical activity and the metabolic syndrome (MetS) or the number pf MetS components (MetScomponents) at different examinations in the a) ULSAM and b) PIVUS studies, and in different age-groups in the c) EpiHealth study.** The regression coefficient (Beta, log odds) vs PA is given. Details are given in S2 Table.

In EpiHealth, a fairly homogenous picture was seen with highly significant negative relationships between PA and MetS or number of MetS components in each of the age-groups. Adjustments were performed for sex, smoking, education level, and alcohol intake.

In ULSAM, similar relationships as described above were seen, with inverse relationships between PA and MetS or number of MetS components. However, in this case, the relationships were strongest at age 70 compared to both younger and older ages.

In PIVUS, similar relationships as described above were seen, with inverse relationships between PA and MetS or number of MetS components. However, in this case, the strength of these relationships declined by age.

The details from the relationships described above are given in S3 Table.

## Longitudinal relationships between PA and MetS

In ULSAM, the change in PA between 50 and 82 years was related to the change in the presence of MetS as well as the change in the number of MetS components, as could be seen in Table 3. Thus, an increase in PA over time was associated with lower risk of obtaining MetS over time. Adjustment was performed for smoking and education.

In PIVUS, the same pattern was seen, but in this case only the change in the number of MetS components was significantly related to the change in PA (p = 0.016), not the change in MetS (p = 0.18).

## Discussion

### Principal findings

Using multiple longitudinal samples, we found that PA increased in men, but not in women, from middle-age to approximately age 70. After age 70, a decline in PA was seen in the elderly.

We furthermore found that a change in PA was related to changes in especially serum triglycerides and HDL-cholesterol, but to a less degree also in fasting glucose and BMI. The associations between PA and CV risk factors were less evident at a high age.

### Change in PA over time

Using longitudinal data, the present study clearly shows a decline in PA from age 70 to older age. This is seen in both sexes and both when the baseline investigation was performed in the 1990-ties or a decade later (ULSAM, PIVUS, SHM).

**Table 3. Relationships between change in presence of the Metabolic syndrome (MetS) or change in number of MetS components and change in physical activity from age 50 to age 82 in ULSAM, and between age 70 and 80 in PIVUS.**

| ULSAM | | | |
|---|---|---|---|
| Variable | Beta (log odds) | SE | p-value |
| MetS | -.31 | .086 | .0004 |
| MetS components | -.17 | .050 | .001 |
| **PIVUS** | | | |
| Variable | Beta (log odds) | SE | p-value |
| MetS | -.23 | .17 | .18 |
| MetS components | -.25 | .10 | .016 |

The study also clearly shows that an improvement in PA over 13–20 years in middle-aged men (ULSAM, SHM) is independent of if the baseline was collected in the 1970s or in the 1990s. In women this pattern was not evident, since a decline was seen in the SHE study between 2001 and 2010, and the interaction between age and time was not significant, indicating that the decline was not restricted to the elderly.

In addition to these longitudinal data, cross-sectional data from the EpiHealth and Life-Gene studies indicated that both sedentary behavior and a high PA was less common in the older age-groups, while moderate PA was most common in the youngest age-groups. PA was generally lower in women compared to men in all age-groups.

These data are well in line with previous literature showing a decline in PA over time being more pronounced in women [12–18]. A novel finding is that PA increased over time in middle-aged subjects up to the age of 70. One reason for this could be that individuals have time to increase their PA following retirement (in Sweden usually at age 65). However, one longitudinal study, which has studied this in detail, could not find such an increase in leisure time PA, even though the job-related PA disappeared at retirement [31].

Of interest is also to notice that in the cross-sectional study (EpiHealth/LifeGene) conducted in 2009–2018, young subjects both reported higher proportions of sedentary behavior as well as higher proportions of sport activities. Thus, in the era of today, both of these extreme parts of the distribution of PA are very common in the young adult generation, while the elderly generally are committed to a modest PA activity. These patterns have not previously been well described.

Trajectories of PA over long time periods within the same adult sample are not commonly described in the literature. We found only one study which combined data from different samples to create trajectories over long time periods [32]. Of interest is that we identified a group of men in ULSAM that started as sedentary, but increased their PA over the first 20 years being around one fourth of the sample, while another fourth were moderately active at middle-age but declined in PA over time. The group with increased PA over time has previously been shown to possess a reduced mortality risk [23].

In one study of middle-aged men followed for 20 years, three trajectories were found; low decreasing, light stable and moderate increasing [33]. This pattern was quite different from those we found in ULSAM over 40 years. No similar trajectories were identified in the elderly PIVUS sample, in which all trajectories declined with ageing, as being in agreement with a previous study in elderly men [34].

## Relationships between PA and CV risk factors

Combining information from the cross-sectional and longitudinal analyses performed in ULSAM and PIVUS shows that PA is most closely related to TG (inverse) and HDL. This picture is more evident in middle-aged subjects than in the elderly. Some evidence of inverse relationships vs BMI and fasting glucose were also found, while relationships vs SBP and LDL are less evident.

The trajectory analysis in ULSAM and PIVUS also supported this view. Of interest is to note that no differences in the group with a low PA at baseline, but increasing during the first 20 years and the group with a moderate PA at baseline, but declining with time was seen regarding risk factor profile. Thus, sedentary subjects can catch up in this respect if they start exercising.

Using cross-sectional data from EpiHealth points towards the same direction, but in this case also BMI and fasting glucose was clearly related to PA. Generally, the relationships were stronger in middle-age than at higher age. The more distinct results and narrower confidence

limits in EpiHealth might be due to the fact that this sample is 4–8 times larger at all ages than ULSAM and EpiHealth.

Intervention trials with increased PA have in meta-analyses shown a better glucose control, improved lipid levels (TG and HDL), and a lower blood pressure [19–21]. Thus, our epidemiological findings are well in line with those data. The novelty in our analysis is that we take age into account and could show that relationships between PA and CV risk factors generally lose strength during ageing. This might be explained by the previous findings that PA declines with ageing and thereby the positive effects of PA on risk factors.

## PA and BMI

The relationships between PA and several risk factors could well be mediated by BMI, since obesity is causally related to many of those [35]. However, the relationship between BMI and PA might either be due to the fact that increasing PA could lower body weight, but also that obese individuals do less exercise. We have therefore evaluated two scenarios, one in which BMI is along the causal pathway between PA and risk factors, and one in which PA is along the causal pathway between BMI and risk factors. In the first scenario, BMI explained 18–20% of the effect of PA on TG or HDL. In the second scenario, PA was not a significant mediator in the BMI vs TG (or HDL) relationship. One way to sort this out is to use Mendelian randomization, but to date the genetic instrument identified for PA explains less than 0.5% of the variation of PA [36], and is therefore too weak to be used in this kind of analyses with sufficient power, as also discussed in the introduction part.

## Strength and limitations

The major strength of the present study is that we were able to use data from several longitudinal studies in both sexes and during different calendar periods. Combining results from those studies gave us a good picture on how PA changes over time in males and females. It also gave us the opportunity to study relationships between PA and CV risk factors in a longitudinal fashion at different ages, which to our knowledge has not been done in the past.

Amongst the limitations is that we only studied Caucasians from one country, which makes the results less easy to generalize.

In the present study, several samples were used and analyzed separately. Since some of the studies only includes one sex, the samples have very different follow-up periods and the samples have used different definitions to grade PA, we have chosen not to pool data from the different samples, but rather regard them as pieces of information that would complement each other.

All other studies except EpiHealth used an overnight fast. In EpiHealth, 6h of fasting was used. The only variable in this that could possibly be affected by this shorter fasting time is triglycerides. Glucose and the other lipids would be back at fasting levels 6h following a meal. Therefore, the results regarding triglycerides might be affected by this shorter fasting time and should therefore be taken with caution.

In the present study, many statistical tests have been performed and using a p-value limit of 0.05 would result in some false positive findings. The majority of the reported associations showed low p-values, but results with p-values close to this limit should be taken with caution if not reproduced in the other studies, such as the mediation analysis for BMI. We did however choose to report the actual p-values not adjusted for multiple testing, since we in most of the cases had more than one study with the same analysis so that validation could be performed.

In the present study the definitions of the different steps on the scale used for PA assessment was not identical between the studies. Although step 1 always denotes a sedentary

lifestyle and the highest step always defines those with highly active training, the steps in between could differ to some extent and therefore might have contributed to any differences seen between the results from different samples.

We were also unable to perform Mendelian randomization with a good power to sort out causality regarding the reported relationships.

In the present study we used self-reported PA to define PA. This is a fairly crude way to evaluate PA compared to direct measurements by accelerometery. However, accelerometery is a rather recently developed technique, so unfortunately we have to wait many years before we have longitudinal studies with repeated accelerometery spanning decades of follow-up.

The present study includes several Swedish studies. This has the advantage that ethnicity, social and other habits are quite uniform and that those factors will not contribute to the total variance in the measured parameters. The disadvantage with a homogenous population is that it is hard to generalize findings to other populations. Therefore, the results of the present study have to be validated in other samples with different ethnicities as well as other geographical locations.

## Conclusion

Using multiple longitudinal samples, we found an increase in PA from middle-age up to 70 years in males, but not in females. Following age 70, a decline in PA was seen. PA was mainly associated with triglycerides and HDL, but also weaker relationships vs fasting glucose, blood pressure, and BMI was found. These relationships were generally less strong in elderly subjects.

## Supporting information

**S1 Table. Definitions of PA in the different cohorts.**
(DOCX)

**S2 Table. Relationships between physical activity and different cardiovascular risk factors at different examinations in the ULSAM and PIVUS studies, and in different age-groups in the EpiHealth study.** The regression coefficient (Beta) vs PA is given.
(DOCX)

**S3 Table. Relationships between physical activity and the metabolic syndrome (MetS) or the number pf MetS components (MetScomponents) at different examinations in the ULSAM and PIVUS studies, and in different age-groups in the EpiHealth study.** The regression coefficient (Beta, log odds) vs PA is given.
(DOCX)

## Acknowledgments

### Disclaimer

Björn Zethelius is employed at the Swedish Medical Products Agency, SE-751 03 Uppsala, Sweden. The views expressed in this paper are the personal views of the authors and not necessarily the views of the Swedish governmental agency.

## Author Contributions

**Conceptualization:** Lars Lind.

**Formal analysis:** Lars Lind.

**Investigation:** Lars Lind, Björn Zethelius, Eva Lindberg, Nancy L. Pedersen, Liisa Byberg.

**Project administration:** Eva Lindberg, Nancy L. Pedersen.

**Writing – original draft:** Lars Lind.

**Writing – review & editing:** Björn Zethelius, Eva Lindberg, Nancy L. Pedersen, Liisa Byberg.

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
