## [Decision Letter · Decision Letter 0]

9 Jun 2021

PONE-D-21-14877

CHANGES IN LEISURE-TIME PHYSICAL ACTIVITY DURING THE ADULT LIFE SPAN AND RELATIONS TO CARDIOVASCULAR RISK FACTORS - RESULTS FROM MULTIPLE SWEDISH STUDIES

PLOS ONE

Dear Dr. Lind,

Thank you for submitting your manuscript to PLOS ONE. After careful consideration, we feel that it has merit but does not fully meet PLOS ONE’s publication criteria as it currently stands. Therefore, we invite you to submit a revised version of the manuscript that addresses the points raised during the review process.

We look forward to receiving your revised manuscript.

Kind regards,

David Meyre

Academic Editor

PLOS ONE

Journal Requirements:

Reviewers' comments:

Reviewer's Responses to Questions

**Comments to the Author**

1. Is the manuscript technically sound, and do the data support the conclusions?

Reviewer #1: No

Reviewer #2: Yes

2. Has the statistical analysis been performed appropriately and rigorously? 

Reviewer #1: No

Reviewer #2: Yes

3. Have the authors made all data underlying the findings in their manuscript fully available?

Reviewer #1: No

Reviewer #2: Yes

4. Is the manuscript presented in an intelligible fashion and written in standard English?

Reviewer #1: Yes

Reviewer #2: Yes

5. Review Comments to the Author

Reviewer #1: PlOS ONE: Changes in leisure-time physical activity during the adult life span and relations to cardiovascular risk factors- results from multiple Swedish studies.

Manuscript number: PONE-D-21-14877.

Please see attachment for review.

Reviewer #2: It is a sound and clearly structured manuscript.

The analysis is done appropriately and based on a large number of individuals.

The meta-analysis is based on a very homogeneous population, several studies being focused on one particular city population (Uppsala). Therefore I find limited interest in the analysis of PA changes according to sex and age, as it is very difficult to extend the findings to other populations with different cultures, living conditions, and social frameworks.

I find that the interest of the study lies mainly on the associations between cardiovascular risk factors and PA levels.

6. PLOS authors have the option to publish the peer review history of their article (what does this mean?). If published, this will include your full peer review and any attached files.

Reviewer #1: No

Reviewer #2: **Yes: **David C. Missud

---

## [Author Response · Author response to Decision Letter 0]

2 Aug 2021

Regarding the editorial comments:

Regarding data availability, we have now Inserted the following statement:

Due to Swedish laws on personal integrity and health data, as well as the Ethics Committee, we are not allowed to make any data included health variables open to the public even if made anonymous. The data could be shared with other researchers after a request to the steering committee (karl.michaelsson@surgsci.uu.se).

My ORCID-ID is supplied.

 

5. Review Comments to the Author

Reviewer #1: PlOS ONE: Changes in leisure-time physical activity during the adult life span and relations to cardiovascular risk factors- results from multiple Swedish studies.

Manuscript number: PONE-D-21-14877.

Please see attachment for review. 

Reviewer #2: It is a sound and clearly structured manuscript.

The analysis is done appropriately and based on a large number of individuals.

The meta-analysis is based on a very homogeneous population, several studies being focused on one particular city population (Uppsala). Therefore I find limited interest in the analysis of PA changes according to sex and age, as it is very difficult to extend the findings to other populations with different cultures, living conditions, and social frameworks.

I find that the interest of the study lies mainly on the associations between cardiovascular risk factors and PA levels.

Reply: We have now added to the limitations section:

“The present study includes several Swedish studies. This has the advantage that ethnicity, social and other habits are quite uniform and that those factors will not contribute to the total variance in the measured parameters. The disadvantage with a homogenous population is that it is hard to generalize findings to other populations. Therefore, the results of the present study have to be validated in other samples with different ethnicities, as well as other geographical locations.”

Detailed comments from reviewer #1:

The present study analyzes trends in leisure time physical activity (LTPA) and cardiovascular disease risk factors from multiple studies in Sweden. The authors identify significant increases in LTPA across the lifespan and demonstrate the LTPA was associated with an improved cardiovascular risk profile.

Minor Comments

Line 28: The authors mention the debate about the optimal diet and alcohol intake for CVD yet this isn’t evaluated in the paper. 

Reply: You are right. We have now dropped that part of the first sentence.

Line 32: Since the authors mention genetic contributors to PA and fitness, the paper could be strengthened by providing heritability estimates for physical activity, fitness and CVD, as well as mentioning any shared heritability. 

Reply: We have now added to the introduction: 

“In a large study including twins in different European countries, the heritability of PA in males and females was similar and ranged from 48% to 71% (6). Such high heritability was confirmed in another twin study (7). The estimated heritability of CVDs, like coronary heart disease is somewhat lower (30-60%) (8).

Two studies have used the Mendelian randomization (MR) approach to see if genes linked to PA also are linked to CVD, thereby suggesting a causal effect of PA on CVD. One of these studies showed a suggestive causal genetic effect of PA on coronary heart disease (9), but not stroke, while no such associations were found in another study using MR (10). It should however be remembered that the number and strength of the SNPs used as genetic instrument for PA is quite low and therefore the risk of false negative findings are high.

Long-term randomized trials with increased PA does not exists, but a community-based intervention study performed in the primary care setting over 9 months showed an improvement in blood pressure and lipids after 9 months, but also a significant reduction in incident CVD after 2 years (11).”

Line 41: The authors state that the links between PA and CVD in different age groups is “poorly studied” yet there is a plethora of literature on this topic. If the authors have reason to think that there are age-dependent effects of PA on CVD risk factors, more evidence should be provided to justify this research question. 

Reply: We have now rephrased that and motivated why we studied if there are different effects of PA on CVD risk factors at different ages:” However, since the impact of these risk factors on future risk for CVD vary with age (22), we hypothesized that also the relationship between PA and CVD risk factors varied in strength during the lifespan. Ideally, also in this case longitudinal data should be used in which the relationship between PA and CVD risk factors should be studied in the same sample at different ages.”

Line 51: Including the sample sizes of each study in the methods would be informative for the reader.

Reply: The sample sizes were given in Table 1, but we have now incorporated this also in the description of the samples.

Line 93: Are there issues with including overnight fasts with a 6 hour fast? If so, a sensitivity analysis should be considered. 

Reply: All other studies except EpiHealth used an overnight fast and since the results from the studies are given separately, we do not see how a sensitivity analysis on this matter should be performed. 

The only variable in this that could possibly be affected by this shorter fasting time is triglycerides. Glucose and the other lipids would be back at fasting levels 6h following a meal.

We have therefore added to the limitation section: “All other studies except EpiHealth used an overnight fast. In EpiHealth, 6h of fasting was used. The only variable in this that could possibly be affected by this shorter fasting time is triglycerides. Glucose and the other lipids would be back at fasting levels 6h following a meal. Therefore, the results regarding triglycerides might be affected by this shorter fasting time and should therefore be taken with caution.”

Major Comments

Overall the introduction needs to be significantly revised to outline the current evidence on the topic and specify the research gap that this study is addressing. 

Reply: We have now made a major revision of the introduction and tried to explain more clearly what this study could contribute with and to more in details specify the objectives. We have also made changes in the discussion accordingly, being more clear on the results in terms of primary and secondary objective. 

The authors outline several statistical tests in each study that adjust for a variety of confounders and use different statistical methods. This makes it unclear to the reader what the primary objectives of the study are and how the results can be compared across studies. Without providing a more succinct analysis, such as performing a pooled analysis, the findings do not sufficiently support the conclusions of the study. In addition, many statistical tests are outlined in the methods without any adjustment for multiple testing.

Reply: Yes we agree that we have used several different statistical test, but it should be noted that we have used the same kind of test for all samples regarding a specific research question. However, since we first have different ages and different sex in the studies, and secondly have measured PA on different scales our statistical advisor STRONGLY advised us not to pool the individual data or perform a meta-analysis, since that approach could be highly misleading. The view of our statistical advisor was that the different samples rather should be used to complement each other to support the conclusion drawn.

We have therefore added a paragraph to the discussion on this issue:” In the present study, several samples were used and analyzed separately. Since some of the studies only includes one sex, the samples have very different follow-up periods and the samples have used different definitions to grade PA, we have chosen not to pool data from the different samples, but rather regard them as pieces of information that would complement each other.”

Regarding the question on p-value and multiple testing, we have consulted our statistician and added a new paragraph in the discussion on this issue: ”In the present study, many statistical tests have been performed and using a p-value limit of 0.05 would result in some false positive findings. The majority of the reported associations showed low p-values, but results with p-values close to this limit should be taken with caution if not reproduced in the other studies, such as the mediation analysis for BMI. We did however choose to report the actual p-values not adjusted for multiple testing, since we in most of the cases had more than one study with the same analysis so that validation could be performed.”

Issues with data harmonization should be discussed. The anchors of each scale are quite different for some measures (e.g., 5= hard exercise > 2 days per week in PIVUS and 5= hard exercise 30 minutes per day in EpiHealth). The implications of this harmonization and potential measurement error should be discussed. 

Reply: As stated above, the differences in the definition of the scales was one of the reasons for not pooling data across studies. We have now added a paragraph on the limitation of using different scales:” In the present study the definitions of the different steps on the scale used for PA assessment was not identical between the studies. Although step 1 always denotes a sedentary lifestyle and the highest step always defines those with highly active training, the steps in between could differ to some extent and therefore might have contributed to any differences seen between the results from different samples.”

---

## [Editor Report · Decision Letter 1]

9 Aug 2021

CHANGES IN LEISURE-TIME PHYSICAL ACTIVITY DURING THE ADULT LIFE SPAN AND RELATIONS TO CARDIOVASCULAR RISK FACTORS - RESULTS FROM MULTIPLE SWEDISH STUDIES

PONE-D-21-14877R1

Dear Dr. Lind,

We’re pleased to inform you that your manuscript has been judged scientifically suitable for publication and will be formally accepted for publication once it meets all outstanding technical requirements.

Kind regards,

David Meyre

Academic Editor

PLOS ONE
---

## [Editor Report · Acceptance letter]

11 Aug 2021

PONE-D-21-14877R1 

Changes in Leisure-Time Physical Activity During the Adult Life Span and Relations to Cardiovascular Risk Factors - Results from Multiple Swedish Studies 

Dear Dr. Lind:

I'm pleased to inform you that your manuscript has been deemed suitable for publication in PLOS ONE. Congratulations! Your manuscript is now with our production department. 

Kind regards, 

on behalf of

Dr. David Meyre 

Academic Editor

PLOS ONE